# TRIM26 Maintains Cell Survival in Response to Oxidative Stress through Regulating DNA Glycosylase Stability

**DOI:** 10.3390/ijms231911613

**Published:** 2022-10-01

**Authors:** Sifaddin M. R. Konis, Jonathan R. Hughes, Jason L. Parsons

**Affiliations:** 1Department of Molecular and Clinical Cancer Medicine, University of Liverpool, 6 West Derby Street, Liverpool L7 8TX, UK; 2Clatterbridge Cancer Centre NHS Foundation Trust, Clatterbridge Road, Bebington CH63 4JY, UK

**Keywords:** DNA damage, DNA repair, OGG1, NEIL1, NTH1, TRIM26, ubiquitin

## Abstract

Oxidative DNA base lesions in DNA are repaired through the base excision repair (BER) pathway, which consequently plays a vital role in the maintenance of genome integrity and in suppressing mutagenesis. 8-oxoguanine DNA glycosylase (OGG1), endonuclease III-like protein 1 (NTH1), and the endonuclease VIII-like proteins 1–3 (NEIL1–3) are the key enzymes that initiate repair through the excision of the oxidized base. We have previously identified that the E3 ubiquitin ligase tripartite motif 26 (TRIM26) controls the cellular response to oxidative stress through regulating both NEIL1 and NTH1, although its potential, broader role in BER is unclear. We now show that TRIM26 is a central player in determining the response to different forms of oxidative stress. Using siRNA-mediated knockdowns, we demonstrate that the resistance of cells to X-ray radiation and hydrogen peroxide generated as a consequence of *trim26* depletion can be reversed through suppression of selective DNA glycosylases. In particular, a knockdown of *neil1* or *ogg1* can enhance sensitivity and DNA repair rates in response to X-rays, whereas a knockdown of *neil1* or *neil3* can produce the same effect in response to hydrogen peroxide. Our study, therefore, highlights the importance of TRIM26 in balancing cellular DNA glycosylase levels required for an efficient BER response.

## 1. Introduction

The base excision repair (BER) pathway plays a vital role in the repair of oxidative DNA base damage and DNA single-strand breaks, and consequently in the maintenance of genome stability. This is exemplified by the fact that cellular DNA is continually subject to reactive oxygen species (ROS) generated by endogenous and exogenous sources, such as through oxidative metabolism and by ionizing radiation, respectively. Estimates of the levels of DNA lesions are ~10,000 per cell per day [1], and if these are not efficiently and effectively repaired, this can lead to mutagenesis and ultimately to the development of human diseases, such as premature ageing, neurodegenerative diseases, and cancer. BER is initiated by one of eleven damage-specific DNA glycosylases [2,3], which act to excise the DNA lesion, and which then stimulates downstream repair activities including AP site incision by AP endonuclease-1, single-nucleotide incorporation by DNA polymerase β, and DNA ligation by the DNA ligase IIIα-X-ray cross-complementing protein 1 complex [4]. The most critical DNA glycosylases that are involved in the recognition and repair of the majority of oxidative DNA base damages are 8-oxoguanine DNA glycosylase (OGG1), endonuclease III-like protein 1 (NTH1; also known as NTHL1), and the endonuclease VIII-like proteins 1–3 (NEIL1–3). However, other DNA glycosylases such as MutY DNA glycosylase (MYH) and thymine DNA glycosylase can also excise oxidized DNA bases. Nevertheless, OGG1 and NTH1 are considered the major enzymes responsible for the repair of 8-oxoguanine and oxidized pyrimidines, respectively [5,6]. In contrast, NEIL1 and NEIL2 appear to have roles in the repair of oxidative lesions in single-stranded DNA [7,8] and, therefore, during transcription and DNA replication. NEIL3 has also been observed to be more active on single-stranded DNA-containing lesions [9], with indications of roles specifically in the repair of interstrand crosslinks and quadruplex DNA [10,11,12,13,14].

It is known that the BER pathway is tightly controlled by individual protein post-translational modifications that act to control the levels, activities, and interactions of the proteins required for an efficient cellular DNA damage response and, therefore, to suppress the accumulation of DNA lesions [15,16]. Protein ubiquitylation, catalyzed ultimately by E3 ubiquitin ligases, has in particular been shown to be a key mechanism through which the cellular protein levels of BER are regulated, both at a steady-state level but also coordinated in response to oxidative stress [17,18]. Approximately 600–700 E3 ubiquitin ligases are present in the human genome, but each have their own target substrate specificity, where they act to transfer ubiquitin moieties predominantly onto specific lysine residues present within the protein. The addition of multiple ubiquitin units through internal lysine residues leading to polyubiquitylation can target the protein for degradation by the 26S proteasome. However, ubiquitylation is a reversible protein post-translational modification that is catalyzed by deubiquitylation enzymes (DUBs) [19]. Our previous evidence has demonstrated that the cellular levels of DNA polymerase β are coordinated by the E3 ubiquitin ligases Mcl-1 ubiquitin ligase E3 (Mule) and C-terminal of Hsc70-interacting protein (CHIP), which is counterbalanced by the DUB-ubiquitin-specific protease 47 (USP47) [20,21,22]. However, we also have more recently demonstrated that Mule can target NEIL1 for ubiquitylation-dependent degradation [23], and so is involved in regulating the two proteins at different steps in the BER process. This demonstrates that the BER response to DNA damage can be efficiently controlled through ubiquitylation driven by a specific set of E3 ubiquitin ligases.

We also recently demonstrated that another E3 ubiquitin ligase, tripartite motif 26 (TRIM26), plays a major role in the regulation of the protein levels of both NEIL1 and NTH1 required for cell survival in response to DNA damage stress [23,24], further demonstrating that a single E3 enzyme can target multiple BER proteins. These studies were performed using different sources of oxidative stress, ionizing radiation and hydrogen peroxide, and so at this stage, it was difficult to understand the contribution of TRIM26-dependent regulation of the two DNA glycosylases under both of these conditions. TRIM26 is one member of the tripartite motif proteins, many of which contain a N-terminal RING finger domain that catalyzes ubiquitylation and has broad cellular roles, including in autophagy and innate immunity, and whose dysregulation is implicated in several cancer types [25,26]. TRIM26 specifically has been shown to be downregulated in hepatocellular carcinoma and papillary thyroid carcinoma [27,28], but upregulated in bladder cancer [29]. Alteration of the expression of *trim26* in cell lines from these tumors was found to impact cell proliferation and migration. In terms of molecular mechanisms and separate from its role in BER, TRIM26 has been demonstrated to target the transcription factor IRF3 for ubiquitylation-dependent degradation, leading to reduced interferon β production and an antiviral response [30]. TRIM26 is also thought to control the inflammatory innate immune response through polyubiquitylation of TAB1 and enhancing NF-κB and MAPK signaling [31]. Moreover, TRIM26 has been indicated in the regulation of ZEB1 protein degradation via ubiquitylation, which alongside USP39, controls the epithelial-to-mesenchymal transition and ultimately the growth of hepatocellular carcinoma cells [32]. It is therefore clear that TRIM26 has multiple cellular targets and roles through its action as an E3 ubiquitin ligase.

Here, we utilized an siRNA knockdown of *trim26* to further define its role in the regulation of the DNA glycosylases required for the response to oxidative stress. Interestingly, we discovered that the increased resistance and DNA repair activity of *trim26*-knockdown cells to ionizing radiation can be suppressed in combination with a knockdown of *neil1* or *ogg1*, whereas this phenotype in response to hydrogen peroxide appears dependent on *neil1* and *neil3*. We also demonstrate that purified TRIM26 can ubiquitylate NEIL1, OGG1, NTH1, and NEIL3 proteins in vitro, suggesting that the enzyme has a wider role in the regulation of DNA glycosylases than previously thought.

## 2. Results

### 2.1. TRIM26 Can Ubiquitylate Multiple Oxidative DNA Glycosylases In Vitro

Our previous evidence has demonstrated that TRIM26 can ubiquitylate both NEIL1 and NTH1 in vitro, which is important in promoting cell survival in response to ionizing radiation and hydrogen peroxide, respectively [23,24]. We, therefore, initially further explored the substrate specificity of purified TRIM26 protein against other DNA glycosylases, using in vitro ubiquitylation assays. We demonstrate that not only can TRIM26 ubiquitylate NEIL1 and NTH1 (Figure 1A,B) in keeping with our previous data, but it can also promote ubiquitylation of OGG1 and NEIL3 (Figure 1C,D). The degree of ubiquitylation efficiency of the DNA glycosylases by TRIM26 varies, and, in some cases (such as NTH1), appears to achieve a level of saturation. Nevertheless, this suggests that at least in vitro, TRIM26 has a broad substrate specificity and can target multiple DNA glycosylases for ubiquitylation.

### 2.2. Acquired Resistance of TRIM26 Knockdown Cells to Ionizing Radiation Can Be Overcome by Targeting NEIL1 and OGG1

We previously showed that *trim26* knockdown U2OS cells are more resistant to the cell-killing effects of ionizing radiation, as a consequence of the accumulation of steady-state levels of NEIL1 protein [23]. To explore this phenotype further, we performed a double knockdown of *neil1*, *nth1*, *ogg1,* or *neil3* along with *trim26* in order to identify specific combinations that led to restoration of cellular radiosensitivity. We first show that we are able to suppress the levels of TRIM26 protein in U2OS cells using a targeted siRNA knockdown compared to a non-targeting (NT) control siRNA (Figure 2A, compare lanes 1 and 2), and that the knockdown efficiency is retained in the various combinations targeting both *trim26* and the DNA glycosylases (Figure 2A, compare lanes 1 and 3–5). Using clonogenic assays, and as expected, we observe a significantly (*p* < 0.03) acquired resistance of cells to X-ray irradiation in the absence of *trim26* compared to the NT control siRNA-treated cells (Figure 2B,C). However, we demonstrate that this radiosensitivity can be restored using a double knockdown of both *trim26* and *neil1,* which is significant from *trim26* knockdown alone (*p* < 0.0001), which highlights the association of the radioresistance of *trim26*-deficient cells with an accumulation of NEIL1 protein, which we previously observed [23]. In contrast, an siRNA knockdown of either *nth1* (Figure 2D,E) or *neil3* (Figure 2F,G) was unable to enhance the radiosensitivity of *trim26*-deficient cells. Surprisingly, we discovered that the combination of *ogg1* and *trim26* siRNA led to radiosensitivity that was significantly different from *trim26* knockdown alone (*p* < 0.002), and similar to that observed in the NT control siRNA-treated cells (Figure 2H,I). This suggests that cellular resistance to ionizing radiation in the absence of *trim26* is dependent on *neil1* and *ogg1*.

### 2.3. TRIM26 Controls the Rates of Repair of Radiation-Induced DNA Damage in a NEIL1 and OGG1-Dependent Manner

To correlate the effects of the DNA glycosylase knockdowns in combination with *trim26* on X-ray-induced cell survival relative to DNA repair, we analyzed the rates of repair of DNA damage in U2OS cells using alkaline comet assays. In the absence of *trim26*, we observed an accelerated rate of repair of alkali-labile sites and DNA single-strand breaks compared to the NT control siRNA-treated cells (Figure 3A), suggesting that accumulating DNA glycosylase levels are responsible for enhanced repair activity. In the absence of both *trim26* and *neil1*, the rates of DNA damage repair were restored similar to that of the NT control siRNA (Figure 3A). However, no changes in the rates of repair of alkali-labile sites and DNA single-strand breaks were observed with the combination of a siRNA knockdown of both *trim26* and *nth1* compared to a *trim26* knockdown alone (Figure 3B). Interestingly, and similar to experiments involving *neil1* siRNA, targeting *ogg1* for an siRNA-mediated knockdown also led to a slower rate of repair of radiation-induced DNA damage compared to the *trim26*-deficient cells (Figure 3C). These effects are consistent with the changes in radiosensitivity of the cells observed above (Figure 2A–I) and show that the increased resistance of *trim26* siRNA knockdown cells is driven through repair coordinated by either NEIL1 or OGG1. To provide supporting evidence for this, we overexpressed *neil1* and *ogg1* individually (Figure 3D,G) and demonstrate that this leads to significantly (*p* < 0.04) enhanced resistance of U2OS cells to X-ray irradiation compared to control-transfected cells (Figure 3E,F). We furthermore show that these cells also harbor increased rates of repair of radiation-induced alkali-labile sites and DNA single-strand breaks compared to control cells (Figure 3H,I), correlating with the increased cellular radioresistance due to the higher expressed levels of NEIL1 or OGG1 protein.

We have previously shown that TRIM26 controls the steady-state levels of NEIL1 protein, and that a *trim26* knockdown generates cellular resistance to X-ray irradiation due to an accumulation of NEIL1 [23]. Given our new data that OGG1 also appears to play a role in radiation-induced DNA damage repair and cellular resistance in a TRIM26-dependent manner, we analyzed DNA glycosylase protein levels by quantitative immunoblotting. In whole cell extracts, we observe that the steady-state levels of both NEIL1 and OGG1 increase by ~2.2 and ~1.8-fold, respectively, in the absence of TRIM26 (Figure 4A,B), whereas protein levels of NTH1 remain unchanged. We also analyzed protein levels in response to X-ray irradiation following biochemical fractionation. In keeping with our previously published data [23], we observe that NEIL1 is present in U2OS cells in a soluble fraction (S) and not strongly bound to chromatin (CB; Figure 4C, compare lanes 1 and 2). We also find that NEIL1 accumulates in response to X-ray irradiation (Figure 4C,E), but that the protein levels are moderately ~1.2-fold higher in *trim26*-siRNA-treated cells, particularly at 0.5–1 h post-irradiation, compared to the NT control siRNA (Figure 4D,E). However, it should be noted that protein levels are normalized relative to their respective control, and that the steady-state levels of NEIL1 are already ~2.2-fold higher in *trim26* knockdown cells. Analysis of NTH1 protein reveals that this is majorly chromatin-bound (Figure 4C, compare lanes 1 and 2) as we previously observed [24], but that there are no substantial differences in protein levels in the presence or absence of *trim26* following irradiation (Figure 4C,D,F). This suggests that NTH1 under these conditions is not regulated in a TRIM26-dependent manner. OGG1 protein is found to be present in both a soluble and chromatin-bound form (Figure 4C, compare lanes 1 and 2), and similar to NEIL1, the levels of the protein are higher in *trim26* knockdown cells, particularly in the soluble fraction at 1–6 h post-irradiation, compared to the NT siRNA control irradiated cells (Figure 4C,D,G,H). Nevertheless again, protein levels are normalized relative to their respective control, and the steady-state levels of OGG1 are already ~1.8-fold higher in *trim26* knockdown cells. Cumulatively, this demonstrates that both NEIL1 and OGG1 protein levels are controlled by TRIM26, which mediates the response to X-ray irradiation.

### 2.4. Acquired Resistance of TRIM26 Knockdown Cells to Hydrogen Peroxide Can Be Overcome by Targeting NEIL1 and NEIL3

We performed a double knockdown of *neil1*, *nth1*, *ogg1,* or *neil3* along with *trim26* in U2OS cells and analyzed the sensitivity in response to oxidative stress induced by hydrogen peroxide compared to a *trim26* knockdown alone. Similar to our previous evidence acquired in HCT116 cells [24], we observed that U2OS cells with a *trim26* siRNA-mediated depletion display a significantly (*p* < 0.004) increased resistance to hydrogen peroxide compared to NT siRNA treated cells (Figure 5A). With a double knockdown of both *trim26* and *neil1*, cellular sensitivity is restored to NT siRNA-treated levels, which was significantly (*p* < 0.0001) different compared to *trim26*-deficient cells (Figure 5A). In contrast, an siRNA knockdown of either *nth1* (Figure 5B) or *ogg1* (Figure 5C) has no significant impact on the resistance of *trim26*-deficient cells to hydrogen peroxide. Interestingly, we discovered that the combination of *neil3* and *trim26* siRNA knockdown led to cellular sensitivity to hydrogen peroxide that was similar to that observed in the NT control siRNA, which was again significantly (*p* < 0.0001) different compared to *trim26*-deficient cells (Figure 5D). To correlate these effects on cell survival following hydrogen peroxide with DNA damage repair, we analyzed the rates of repair of alkali-labile sites and DNA single-strand breaks in U2OS cells with the various siRNA knockdown combinations. As expected in the absence of *trim26*, there was an accelerated rate of repair of the DNA damage compared to the NT-control-siRNA-treated cells, but which could be suppressed in combination with a knockdown of *neil1* (Figure 5E). In the absence of both *trim26* and *nth1* (Figure 5F) or of *trim26* and *ogg1* (Figure 5G), the kinetics of DNA damage repair were similar to that of the *trim26*-siRNA-treated-only cells. Targeting *neil3* for an siRNA-mediated knockdown in *trim26*-depleted cells, similar to *neil1*, led to a slower rate of repair of DNA damage induced by hydrogen peroxide (Figure 5H). Effects on DNA repair rates are consistent with the changes observed in cellular sensitivity (Figure 5A–D), and reflect that *neil1* and *neil3* are the major drivers of increased resistance in *trim26* siRNA knockdown cells. Additional support for this, at least focused on *neil1*, is provided by our observations that NEIL1 overexpression leads to significantly (*p* < 0.02) enhanced resistance of U2OS cells to hydrogen peroxide compared to control-transfected cells (Figure 5I), and that there are also associated increases in the kinetics of repair of alkali-labile sites and DNA single-strand breaks under these conditions (Figure 5J).

We analyzed the endogenous protein levels of NEIL1, NTH1, and NEIL3 in *trim26* knockdown compared to NT control siRNA-treated cells following hydrogen peroxide treatment. The levels of NEIL1 protein within the soluble fraction were ~1.4–1.5-fold higher in *trim26* siRNA-treated cells at 1–6 h post-treatment compared to the NT control siRNA cells (Figure 6A–C). No significant differences in chromatin bound NTH1 protein levels in the presence or absence of *trim26* following treatment were found (Figure 6A,B,D). Similar to NEIL1, NEIL3 protein was observed to be largely present within the soluble fraction and not chromatin bound but also the levels of the protein were ~1.5–1.7-fold higher in *trim26* knockdown cells at 2–6 h post-treatment with hydrogen peroxide compared to the NT siRNA control cells (Figure 6A,B,E). These data indicate that both NEIL1 and NEIL3 protein levels are tightly controlled by TRIM26, which mediates the response to oxidative stress induced by hydrogen peroxide.

## 3. Discussion

BER is an essential DNA repair pathway that responds to cellular oxidative stress and is critical in maintaining genome stability and in preventing mutagenesis. Within this pathway, OGG1, NTH1, and NEIL1–3 are the principal DNA glycosylases that recognize and excise oxidative DNA base lesions, which then promotes subsequent repair coordinated by AP endonuclease-1, DNA polymerase β, and DNA ligase IIIα-X-ray cross-complementing protein 1 complex. A number of studies have demonstrated that the efficiency of the BER pathway is subject to tight control by post-translational modifications, of which ubiquitylation as a mechanism for regulating individual repair protein levels has been increasingly found [15,16]. DNA glycosylases specifically are a target for regulation at both the transcriptional and post-translational level [33], which avoids the build-up of potentially more toxic BER intermediates. The importance of controlling DNA glycosylase levels is displayed by the altered protein expression observed in several diseases, including neurodegenerative diseases and cancer. For example, an altered expression and activity of OGG1 has been observed in head and neck cancers [34,35], of NTH1 in gastric cancer [36], and of NEIL3 in various human cancers [37,38], whereas a loss of NEIL1 causes memory and brain defects indicative of early-onset neurodegenerative disease, similar to those observed in Alzheimer’s and Parkinson’s diseases [39,40]. We previously identified that the E3 ubiquitin ligase TRIM26 can regulate the cellular protein levels of NEIL1 and NTH1 in response to X-ray radiation and hydrogen peroxide-induced stress, respectively [23,24]. In this study, we now provide evidence that TRIM26 can also ubiquitylate OGG1 and NEIL3 in vitro, but that it controls the different DNA glycosylases based on the form of DNA damage stress. Specifically, NEIL1 and OGG1 are responsive to X-ray radiation in a TRIM26-dependent manner, whereas NEIL1 and NEIL3 respond following treatment with hydrogen peroxide.

OGG1 and NTH1 are well established as being the major DNA glycosylases involved in the repair of 8-oxoguanine and oxidized pyrimidines, respectively. In contrast, the NEIL glycosylases appear to have more defined cellular roles, such as in transcription and replication as a consequence of their activity on DNA lesions within single-stranded DNA [3], and in DNA crosslink repair [41]. Therefore, it is understandable that their regulation may be responsive to different types and sources of DNA damage. Interestingly, we observed that the control of NEIL1 is important for the response to both X-rays and hydrogen peroxide in terms of promoting survival and efficient DNA damage repair, indicating that a common DNA lesion is being generated by these sources of oxidative stress that is highly dependent on NEIL1. However, maintenance of OGG1 by TRIM26 only occurs following X-ray irradiation, whereas NEIL3 is tightly controlled by TRIM26 in response to hydrogen peroxide. The reasoning for this is currently unclear but suggests a different DNA lesion dependence that requires either OGG1 or NEIL3 for repair. Interestingly, recent evidence has suggested that 8-oxoguanine may act as a transcriptional regulator and can negatively affect gene transcription when in non-transcribed DNA, but can alternatively promote gene expression, such as when present within a G-quadruplex sequence [42,43,44]. It is therefore tempting to speculate that the levels of OGG1, and possibly NEIL1 and NEIL3, are differentially regulated not only for maintaining 8-oxoguanine throughout the genome, but specifically for its roles in DNA transcription and epigenetic regulation [45,46]. However, this requires more detailed investigation.

In addition to the selective control of the DNA glycosylase levels by TRIM26 relative to the DNA damage stress, an unanswered question is how the mechanism is coordinated. Predictably, this could also be achieved at the post-translational level, either through a competing DUB that is able to reverse the effects of TRIM26-dependent ubiquitylation and degradation of the DNA glycosylase, or through an alternative post-translational modification that either stimulates or inhibits TRIM26 activity. In support of the former, we have previously identified that the DUB USP47 can control the protein levels of DNA polymerase β and provides competition for ubiquitylation catalyzed by CHIP and Mule [20], although no DUBs for the DNA glycosylases OGG1, NTH1, NEIL1, and NEIL3 have yet been identified. In terms of alternative post-translational modifications, OGG1 has been previously demonstrated to be subject to acetylation [47] and phosphorylation [48], which could interfere with ubiquitylation. Similarly, NEIL1 is reportedly phosphorylated [49] and acetylated [50]. However, to our knowledge, no post-translational modifications of NTH1 and NEIL3 have yet been identified. Furthermore, poly(ADP-ribosyl)ation catalyzed predominantly poly(ADP-ribose) polymerase-1 (PARP-1) plays a critical role in coordinating BER, and, thus, in controlling cell survival in response to genotoxic stress [4]. Therefore it could be speculated that this post-translational modification may also have an underlying role in regulating DNA glycosylase stability. Nevertheless, further research needs to be established in order to examine any potential crosstalk between TRIM26-dependent ubiquitylation and other post-translational modifications of the DNA glycosylases. Despite this, our research highlights a central role for TRIM26 in controlling and coordinating the cellular response to DNA damage through DNA glycosylase modulation.

## 4. Materials and Methods

### 4.1. Reagents

NEIL1 antibodies were kindly provided by Dr. T. Rosenquist. Antibodies against TRIM26 (ab89290), NTH1 (ab70726), OGG1 (ab124741), and fibrillarin (ab4566) were from Abcam (Cambridge, UK). NEIL3 (sc-393703) and lamin a/c antibodies (sc-7292) were from Santa Cruz Biotechnology (Dallas, TX, USA), and tubulin antibodies (T6199) were from Merck (Gillingham, UK). Bacterial expression plasmids and protein purification of TRIM26, OGG1, NTH1, NEIL1, and NEIL3 proteins was performed as previously described [23,24,51].

### 4.2. Cell Culture, siRNA Knockdowns, and Plasmid Overexpressions

U2OS cells were cultured at 37 °C in 5% CO_2_ in high-glucose Dulbecco’s modified Eagle’s medium (DMEM) containing 10% fetal bovine serum, 2 mM L-glutamine, 1 × penicillin-streptomycin, and 1 × non-essential amino acids. Cells were authenticated using short-tandem-repeat (STR) profiling and were routinely tested to ensure the absence of mycoplasma infection. To perform siRNA knockdowns, cells were cultured in 35 mm dishes for 24 h to 30–50% confluence and then treated with 2 µL of Lipofectamine RNAiMAX transfection reagent (Life Technologies, Paisley, UK) in the presence of either 80 nM (NT, TRIM26, NEIL1, and NEIL3) or 160 nM (OGG1) siRNA for an additional 72 h. The following siRNA sequences were used: Qiagen AllStars Negative Control siRNA (Qiagen, Southampton, UK), TRIM26 siRNA (5′-CCGGAGAAUUCUCAGAUAA-3′), or the appropriate ON-TARGETplus siRNA pools against OGG1, NTH1, NEIL1, or NEIL3 (Horizon Discovery, Cambridge, UK). For overexpression of NEIL and OGG1, 0.2 µg of pCMV-Tag3a mammalian expression plasmids (as previously described [24,51]) was similarly transfected into cells using Lipofectamine 2000 transfection reagent (Life Technologies, Paisley, UK) for 24 h prior to subsequent analysis. Control samples for overexpression were treated with transfection reagent only.

### 4.3. Cell Treatments and Clonogenic Assays

Cells cultured in 35 mm dishes were treated with 1–4 Gy X-rays using the 130 MeV CellRad X-ray irradiator (Faxitron Bioptics, Tucson, AZ, USA), or with 250–1000 µM hydrogen peroxide for 15 min. Cells were washed with PBS, trypsinized, counted, and a defined number seeded in triplicate into 6-well plates. Cells were incubated at 37 °C in 5% CO_2_ for 9 days to promote colony growth, and these were then fixed and stained with 6% glutaraldehyde and 0.5% crystal violet for 30 min. Plates were washed, left to air-dry overnight, and colonies counted using the GelCount colony analyzer (Oxford Optronics, Oxford, UK). The surviving fraction was determined using the number of colonies per treatment level versus the number of colonies achieved in the untreated control. Statistical analysis of the differences across the treatment doses comparing the various gene knockdowns/overexpressions was performed using the CFAssay for R package [52].

### 4.4. Whole-Cell Extract Preparation and Cell Fractionation

Cells were washed, harvested in ice-cold PBS, and whole-cell extracts prepared as previously described [53]. Alternatively, biochemical fractionation was performed immediately to generate soluble and chromatin-bound protein fractions as previously described [23]. In brief, cell pellets were resuspended in two packed cell volumes (PCVs) of buffer containing 20 mM Tris–HCl (pH 7.8), 2.5 mM MgCl2, 0.5% (*v/v*) IGEPAL CA-630, 100 µM PMSF, 1 mM N-ethylmaleimide (NEM), and 1 µg/mL of the protease inhibitors (leupeptin, aprotinin, chymostatin, and pepstatin), and incubated for 10 min on ice. Extracts were centrifuged at 10,000 rpm for 2 min at 4 °C and the supernatant containing soluble proteins (S) was collected. The nuclear pellet was similarly extracted with two PCVs of buffer containing 20 mM NaPO_4_ (pH 8.0), 0.5 M NaCl, 1 mM EDTA, 0.75% (*v/v*) Triton X-100, 10% (*v/v*) glycerol, 100 µM PMSF, 1 mM NEM, and 1 µg/mL of each protease inhibitor and incubated on ice for 10 min. Following centrifugation, the supernatant containing chromatin-bound proteins (CB) was collected. For immunoblotting analysis, 40–70 µg of protein from the S fraction and the same corresponding volume of the CB fraction were used, and proteins were visualized and quantified using the Odyssey image analysis system (Li-cor Biosciences, Cambridge, UK).

### 4.5. In Vitro Ubiquitylation Assay

Ubiquitylation reactions were performed as previously described [23,24,51]. Briefly, reactions containing either histagged-OGG1 (5.2 pmol), NTH1 (5.8 pmol), NEIL1 (4.6 pmol), NEIL3 (3 pmol), and/or TRIM26 (11–22 pmol) were incubated with 0.7 pmol GST-E1 activating enzyme, and 2.5 pmol H5a, H5b, and H5c E2-conjugating enzymes; 0.6 nmol ubiquitin in buffer containing 25 mM Tris-HCl (pH 8.0), 4 mM ATP, 5 mM MgCl_2_, 200 µM CaCl_2_, and 1 mM DTT were prepared and incubated in LoBind protein tubes (Eppendorf, Stevenage, UK) for 1 h at 30 °C with agitation. After terminating the reactions through the addition of SDS-PAGE sample buffer (25 mM Tris-HCl (pH 6.8), 2.5% β-mercaptoethanol, 1% SDS, 10% glycerol, 1 mM EDTA, and 0.05 mg/mL of bromophenol blue), these were heated for 5 min at 95 °C and analyzed by SDS-PAGE and immunoblotting.

### 4.6. Alkaline Single-Cell Gel Electrophoresis (Comet) Assay

The alkaline comet assay was performed as previously described, utilizing in-gel DNA repair activities [54]. In brief, cells were trypsinized, diluted to ~1 × 10^5^ cell/mL, and 250 µL aliquots placed in the wells of a 24-well plate on ice. Following X-ray irradiation (1.5 Gy) or treatment with hydrogen peroxide (10 µM) for 5 min, cells were embedded in 1% low-melting-point agarose (Bio-Rad, Hemel Hempstead, UK), which was added to a microscope slide precoated and dried with 1% normal-melting-point agarose. The agarose was then allowed to set for 2–3 min on ice, and then placed in a humidified chamber for up to 120 min to stimulate DNA repair. Following this, cell lysis was performed by placing slides in 2.5 M NaCl, 100 mM EDTA, 10 mM Tris-HCl, pH 10.5, 1% (*v/v* dimethyl sulfoxide (DMSO), and 1% (*v/v*) Triton X-100 for at least 1 h at 4 °C. Slides were transferred to a darkened comet assay tank (Appleton Woods, Birmingham, UK), incubated for 30 min in fresh cold electrophoresis buffer (300 mM NaOH, 1 mM EDTA, and 1% (*v*/*v*) DMSO, pH 13) to allow the DNA to unwind, and electrophoresis was performed at 25 V, 300 mA for 25 min. Slides were carefully removed from the tank and neutralized three times with 5 min washes of 0.5 M Tris-HCl (pH 8.0), prior to air-drying overnight. Following rehydration of the slides for 30 min in water (pH 8.0), the DNA was stained using SYBR Gold (Life Technologies, Paisley, UK) diluted 1:20,000 in water (pH 8.0) for 30 min, and then again allowed to air-dry. For imaging, cells (50 per slide, 2 slides per time point) were analyzed using the Komet 6.0 image analysis software (Andor Technology, Belfast, Northern Ireland) and average % tail DNA values were determined from three independent, biological experiments.

## Figures and Tables

**Figure 1 ijms-23-11613-f001:**
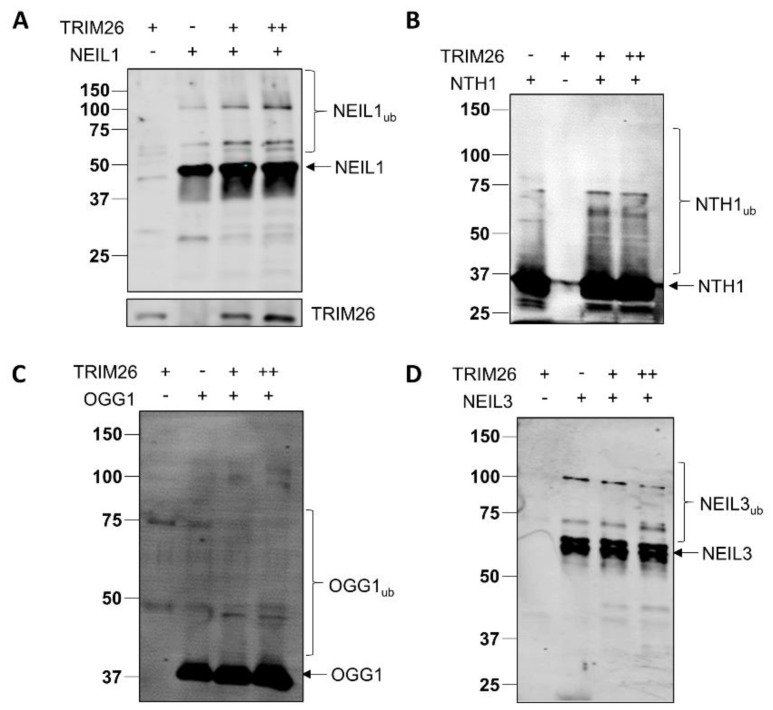
TRIM26 can ubiquitylate NEIL1, NTH1, OGG1, and NEIL3 in vitro. In vitro ubiquitylation of (**A**) His-tagged NEIL1 (4.6 pmol), (**B**) His-tagged NTH1 (5.8 pmol), (**C**) His-tagged OGG1 (5.2 pmol), and (**D**) His-tagged NEIL3 (3 pmol) by His-tagged TRIM26 (11 or 22 pmol). All in vitro ubiquitylation reactions were performed in the presence of E1 activating enzyme (0.7 pmol), H5a, H5b, and H5c E2-conjugating enzymes (2.5 pmol), and ubiquitin (0.6 nmol), and were subsequently analyzed by SDS-PAGE and immunoblotting using the respective antibodies. Molecular-weight markers are indicated on the left-hand side of the blots, and the positions of unmodified and ubiquitylated proteins (e.g., NEIL1_ub_) are displayed.

**Figure 2 ijms-23-11613-f002:**
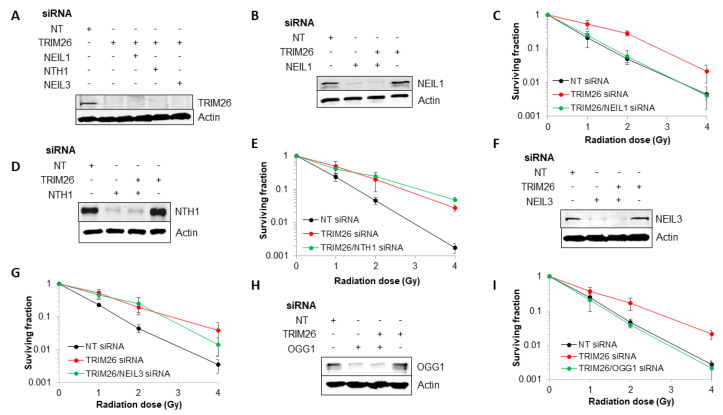
NEIL1 and OGG1 control cell survival in response to X-ray radiation in a TRIM26-dependent manner. (**A**,**B**,**D**,**F**,**H**) WCE from U2OS cells treated with NT control siRNA, plus the various combinations of *trim26* knockdown alone and with DNA glycosylase (*neil1*, *nth1*, *ogg1,* or *neil3*) were prepared and analyzed by SDS-PAGE and immunoblotting using the indicated antibodies. (**C**,**E**,**G**,**I**) Clonogenic survival of cells was analyzed following treatment with increasing doses of X-ray irradiation (0–4 Gy). Shown is the mean surviving fraction ± S.E. from at least three independent experiments.

**Figure 3 ijms-23-11613-f003:**
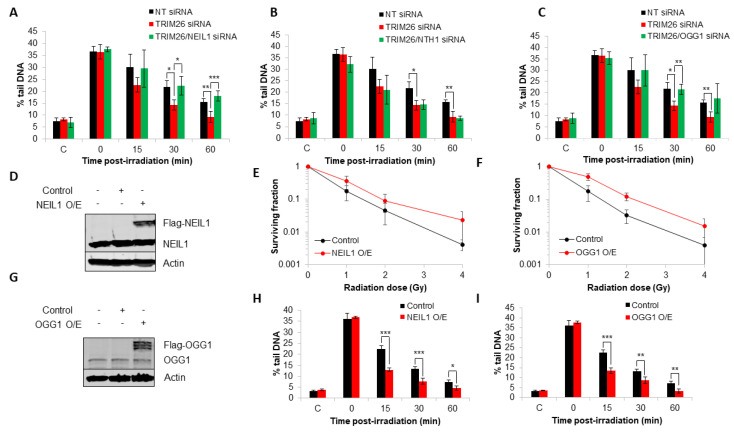
NEIL1 and OGG1 control the repair of DNA damage in response to X-ray radiation in a TRIM26-dependent manner. (**A**–**C**) U2OS cells treated with NT control siRNA, plus the combinations of *trim26* knockdown alone and with either (**A**) *neil1*, (**B**) *nth1*, or (**C**) *ogg1* siRNA were irradiated with X-ray irradiation (1.5 Gy) and alkali-labile sites and DNA single-strand breaks measured at 0–60 min post-treatment using the alkaline comet assay. Shown is the mean % tail DNA ± S.D. * *p* < 0.05, ** *p* < 0.02, and *** *p* < 0.01 as analyzed by a two-sample *t*-test. (**D**,**G**) WCE from U2OS cells with either (**D**) Flag-NEIL1 or (**G**) Flag-OGG1 overexpression were prepared and analyzed by SDS-PAGE and immunoblotting with the indicated antibodies. (**E**,**F**) Clonogenic survival of cells containing either (**E**) NEIL1 or (**F**) OGG1 overexpression was analyzed following treatment with increasing doses of X-ray irradiation (0–4 Gy). Shown is the mean surviving fraction ± S.E from at least three independent experiments. (**H**,**I**) U2OS cells containing either (**H**) NEIL1 or (**I**) OGG1 overexpression were irradiated with X-ray irradiation (1.5 Gy) and alkali-labile sites and DNA single-strand breaks measured at 0–60 min post-treatment using the alkaline comet assay. Shown is the mean % tail DNA ± S.D. * *p* < 0.05, ** *p* < 0.02, and *** *p* < 0.005 as analyzed by a two-sample *t*-test.

**Figure 4 ijms-23-11613-f004:**
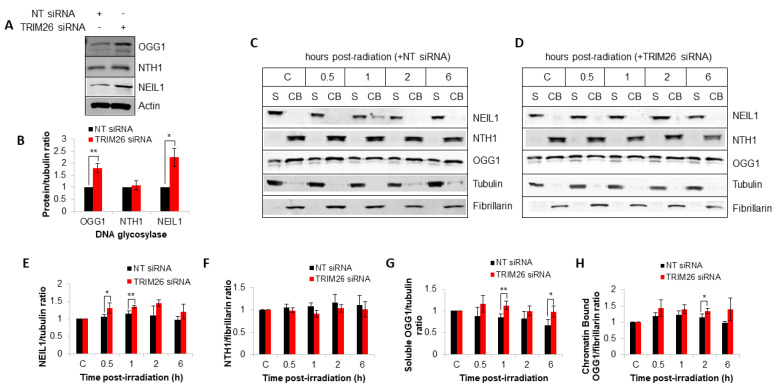
TRIM26 controls steady-state NEIL1 and OGG1 protein levels and those in response to X-ray irradiation. (**A**) U2OS cells treated with NT control or *trim26* siRNA were harvested and whole-cell extracts analyzed by SDS-PAGE and immunoblotting. (**B**) DNA glycosylase levels relative to tubulin are shown as mean ± S.E. * *p* < 0.05 and ** *p* < 0.02, as analyzed by a two-sample *t*-test. U2OS cells treated with (**C**) NT control siRNA or (**D**) *trim26* siRNA were either unirradiated (designated C) or treated with X-ray irradiation (10 Gy) and harvested at the indicated time points post-treatment. Proteins were fractionated into soluble (S) and chromatin-bound (CB) fractions, and analyzed by SDS-PAGE and immunoblotting. Protein levels of (**E**) NEIL1 relative to tubulin, (**F**) NTH1 relative to fibrillarin, (**G**) OGG1 in the S fraction relative to tubulin, and (**H**) OGG1 in the CB fraction relative to fibrillarin (all mean ± S.D.) were quantified from at least three independent experiments and were normalized relative to the respective unirradiated cells, which was set to 1.0. * *p* < 0.05 and ** *p* < 0.02 as analyzed by a two-sample *t*-test.

**Figure 5 ijms-23-11613-f005:**
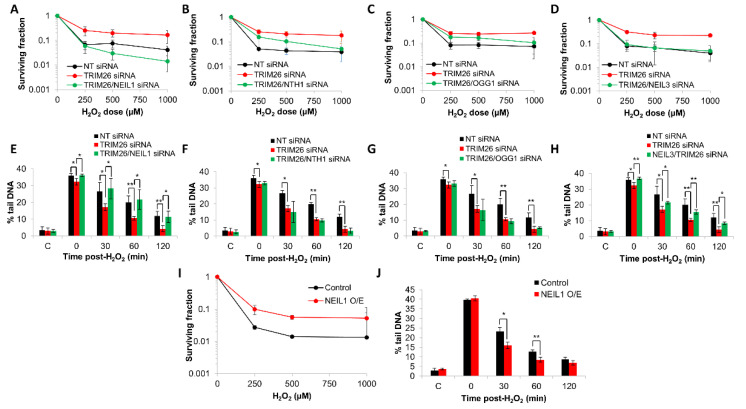
NEIL1 and NEIL3 control cell survival and repair of DNA damage in response to hydrogen peroxide in a TRIM26-dependent manner. (**A**–**H**) U2OS cells were treated with NT control siRNA, plus the various combinations of *trim26* knockdown alone and with the DNA glycosylase (*neil1*, *nth1*, *ogg1*, or *neil3*). (**A**–**D**) Clonogenic survival of cells was analyzed following treatment with increasing doses of hydrogen peroxide (0–1000 µM). Shown is the mean surviving fraction ± S.E from at least three independent experiments. (**E**–**H**) Alternatively, cells were treated with hydrogen peroxide (10 µM) and alkali-labile sites and DNA single-strand breaks measured at 0–120 min post-treatment using the alkaline comet assay. Shown is the mean % tail DNA ± S.D. * *p* < 0.05 and ** *p* < 0.02 as analyzed by a two-sample *t*-test. (**I**) Clonogenic survival of cells containing NEIL1 overexpression was analyzed following treatment with increasing doses of hydrogen peroxide (0–1000 µM). Shown is the mean surviving fraction ± S.E from at least three independent experiments. (**J**) U2OS cells containing NEIL1 overexpression were treated with hydrogen peroxide (10 µM) and alkali-labile sites and DNA single-strand breaks measured at 0–120 min post-treatment using the alkaline comet assay. Shown is the mean % tail DNA ± S.D. * *p* < 0.01 and ** *p* < 0.02 as analyzed by a two-sample *t*-test.

**Figure 6 ijms-23-11613-f006:**
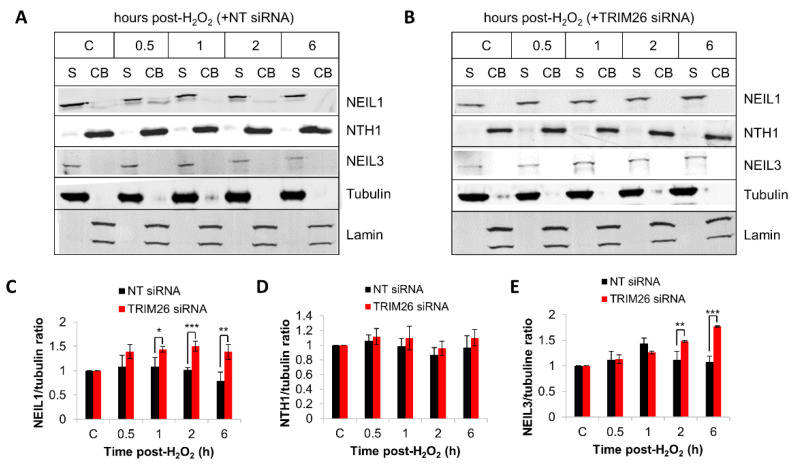
TRIM26 controls NEIL1 and NEIL3 protein levels in response to hydrogen peroxide. U2OS cells treated with (**A**) NT control siRNA or (**Β**) *trim26* siRNA were either untreated (designated C) or treated with hydrogen peroxide (150 µM) and harvested at the indicated time points post-treatment. Proteins were fractionated into soluble (S) and chromatin-bound (CB) fractions, and analyzed by SDS-PAGE and immunoblotting. Protein levels of (**C**) NEIL1 relative to tubulin, (**D**) NTH1 relative to lamin, and (**E**) NEIL3 relative to tubulin (all mean ± S.D.) were quantified from at least three independent experiments, and were normalized relative to the respective unirradiated cells, which was set to 1.0. * *p* < 0.05, ** *p* < 0.02, and *** *p* < 0.002 as analyzed by a two-sample *t*-test.

## Data Availability

The data presented in this study are available on request from the corresponding author.

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
