# Peer review of "TRIM26 Maintains Cell Survival in Response to Oxidative Stress through Regulating DNA Glycosylase Stability"

_ijms, 2022, doi:10.3390/ijms231911613_

Round 1

Reviewer 1 Report

The manuscript by Sifaddin Konis et al., “TRIM26 maintains cell survival in response to ROS via regulating DNA glycosylase stability” describes the role of the E3 ubiquitin ligase tripartite motif 26 (TRIM26) in regulating the cellular level of human oxidative DNA glycosylases: NTHL1, OGG1, NEIL1 and NEIL3. Authors demonstrated that siRNA-mediated downregulation of TRIM26 makes cells more resistant to x-ray radiation and hydrogen peroxide. Interestingly, downregulation of NEIL1 and OGG1 reverse effect of TRIM26 depletion making cells more sensitive and activating DNA excision in response to x-rays, whereas, downregulation of NEIL1 and NEIL3 in TRIM26 depleted cells enhance sensitivity and DNA excision in response to hydrogen peroxide. Authors propose that TRIM26 controls DNA glycosylases in order to avoid the generation of potentially toxic BER intermediates such as DNA strand breaks. Overall, the work is a continuation of authors’ previous studies on the role of TRIM26 in regulating NTHL1 and NEIL1 protein levels. This study reports interesting findings that extend our understanding of how human DNA glycosylases can be regulated in response to genotoxic stress. I have some comments that would require consideration.

General remarks.

1) Authors should mention other human DNA glycosylases that might play critical role in repair of the majority of oxidative DNA base damage. For example, MUTYH excises 2-oxoadenine; whereas mismatch-specific Thymine DNA glycosylase, TDG excises thymine glycol and 8-oxoadenine; in addition human alkyl-purine DNA glycosylase AAG/MPG and TDG can excise exocyclic etheno-DNA adducts that are generated by product of lipid peroxidation.

 2. Authors examined cellular sensitivity and DNA repair activities in cells depleted for TRIM26 or both TRIM26 and a DNA glycosylase. It would be important to examine cellular response in cells depleted for a DNA glycosylase only. Previous studies with DNA glycosylases knockout mice did not reveal increased sensitivity to oxidative stress as compared to wild type strains (Friedberg and Meira, DNA repair, 2006).

 3. Authors measured the rates of repair by using alkaline comet assays. In the absence of TRIM26 authors observed significant decrease in the level of alkali-labile sites and DNA single strand breaks compared to control siRNA-treated cells (Fig. 3), suggesting that increased level of the DNA glycoslyase stimulates efficient repair. It would be interesting to examine the rate of poly(ADP-ribosy)lation in response to x-rays and H2O2. Since the level of poly(ADP-ribosy)lation determines the cellular fate in response to genotoxic stress and offers more precise instrument to study the role of TRIM26 and DNA glycosylases in the repair of oxidative DNA damage.

Minor comment

 4. The generally accepted nomenclature of human endonuclease III is NTHL1, this should be mentioned.

Reviewer 2 Report

In the manuscript titled” TRIM26 maintains cell survival in response to oxidative stress through regulating DNA glycosylase stability”, the authors investigate the role of TRIM26 and DNA glycosylases in response to oxidative stress. Following suggestions should be taken into consideration to improve the quality of the manuscript.

1.     Figure 1. The authors have used increasing concentration of TRIM26 in in vitro ubiquitylation assay. Its not clear from the figures if they depict increasing in the ubiquitylated NTH1, NEIL3 and OGG1. The authors should include a reason for this observation in text.

2.     Can the authors also include a western blot depicting expression of TRIM26?

3.     In Figure 2 and for western blot other figures, the authors should include siRNA in the figure panels as the gene name looks misleading.

4.     In Figure 2, when performing double knockdown with trim26 and NEIL1/NTH1/OGG1, what concentrations of NT siRNA were used, the authors should include that in the methods section.

5.     In Figure 3, When using overexpression systems, did the authors also conduct the experiments using a vector only (flag vector) control?

6.     In Figure 4, the authors depict that the TRIM26 regulates the protein levels of NEIL1 and OGG1. Overexpression of NEIL1 and OGG1 in trim26 knockdown cells could depict the role of OGG1 and NEIL1 in DNA damage response in a better manner as compared to overexpressing in control cells.

7.     In Figure 5A, the authors mention that the TIRM26/NEIL1 double knockdown restores the cellular sensitivity to NT siRNA treated levels. This is true for the 250um dose of H2O2 but at higher dose the cellular sensitivity to H2O2 increases as compared to NT control siRNA. Can the authors explain this observation?
